# Tight bounds for the median of a gamma distribution

**Richard F. Lyon** *

Google Research, Google Inc., Mountain View, California, United States of America

* dicklyon@acm.org

**Data Availability Statement:** All relevant data are within the paper.

**Funding:** The author(s) received no specific funding for this work. Author RFL is employed by Google. The funder provided support in the form of

## Abstract

The median of a standard gamma distribution, as a function of its shape parameter $k$, has no known representation in terms of elementary functions. In this work we prove the tightest upper and lower bounds of the form $2^{-1/k}(A + k)$: an upper bound with $A = e^{-\gamma}$ (with $\gamma$ being the Euler–Mascheroni constant) and a lower bound with $A = \log(2) - \frac{1}{3}$. These bounds are valid over the entire domain of $k > 0$, staying between 48 and 55 percentile. We derive and prove several other new tight bounds in support of the proofs.

## Introduction

We prove some of the bounds conjectured by Lyon [1] for the median of a gamma distribution, relying on previously known and new bounds that are tighter in some regions of the domain, and also relying on numerically transparent evaluations of the CDF at a few points, using a convergent series.

The gamma distribution's probability density function (PDF) is $x^{k-1} e^{-x/\theta}/\Gamma(k)\theta^k$, but we'll use $\theta = 1$ because both the mean and median simply scale with this parameter. Thus we use this "standard gamma distribution" PDF with just the shape parameter $k$, with $k > 0$ and $x > 0$:

$$p_k(x) = \frac{1}{\Gamma(k)} x^{k-1} e^{-x}.$$

The mean $\mu$ of this distribution is well known to be $\mu(k) = k$, which is easy to verify since the first moment of the PDF evaluates to $\Gamma(k + 1)/\Gamma(k) = k$. The median $v(k)$ is the value of $x$ at which the cumulative distribution function (CDF) equals one-half:

$$\frac{1}{2} = \int_0^{v(k)} p_k(x)dx = \frac{1}{\Gamma(k)} \int_0^{v(k)} x^{k-1}e^{-x}dx.$$

We seek to prove closed-form tight bounds for $v(k)$ that achieve 50th percentile in the limit for $k \to 0$ and for $k \to \infty$. By "tight" we mean a bound is equal to the true value at a point, or in the limit at 0 or $\infty$. Informally, there are degrees of tightness, based on how many derivatives are also matched; so one tight bound might be tighter than another.

See Fig 1 for some known linear and piecewise-linear bounds, illustrating the lack of good known bounds for $0 < k < 1$. Many prior publications have specifically only considered $k \geq 1$

salary for RFL and the publication fee for this article, but did not have any additional role in the study design, data collection and analysis, decision to publish, or preparation of the manuscript. There was no additional external funding received for this study. The specific roles of these authors are articulated in the 'author contributions' section. The funders had no role in study design, data collection and analysis, decision to publish, or preparation of the manuscript.

**Competing interests:** Author RFL is employed by Google. This does not alter our adherence to PLOS ONE policies on sharing data and materials. Google has no restrictions on this work.

[2, 3], or only positive integer values [4], or positive half-integers for the Chi-square distribution [5]; the full range $k > 0$ has also been considered [6, 7], but the bounds found leave room for improvement, particularly at low $k$.

## Theorems to prove

We will show that, for all $0 < k < \infty$, the median of the gamma distribution is bounded above and below by:

$$2^{-1/k}(A_L + k) < v(k) < 2^{-1/k}(A_U + k)$$

with closed-form scalar constants $A_L = \log(2) - \frac{1}{3}$ and $A_U = e^{-\gamma}$ (with $\gamma \approx 0.5772157$ being the Euler–Mascheroni constant), and that these bounds are asymptotically tight for $k \to \infty$ and $k \to 0$, respectively. Equivalently, we define the function $A(k)$ and tightly bound it with these constants:

$$A(k) = 2^{1/k}v(k) - k$$

$$\Rightarrow \quad \log(2) - \frac{1}{3} < A(k) < e^{-\gamma}.$$

In addition, we'd like to prove that $A(k)$ is monotonically decreasing between its low-$k$ and high-$k$ limits, as suggested by asymptotic and graphical numerical observations. If we could prove monotonicity, the other proof would be easier, but we don't see how yet. So, the strategy here is to show that over various subsets of the $k$ domain, with their union covering $0 < k < \infty$, there are other bounds that we can prove are between our new closed-form bounds and the true median. Therefore, along the way, we derive several other new upper and lower bounds that are tighter over portions of the $k$ domain, and some asymptotic values and slopes.

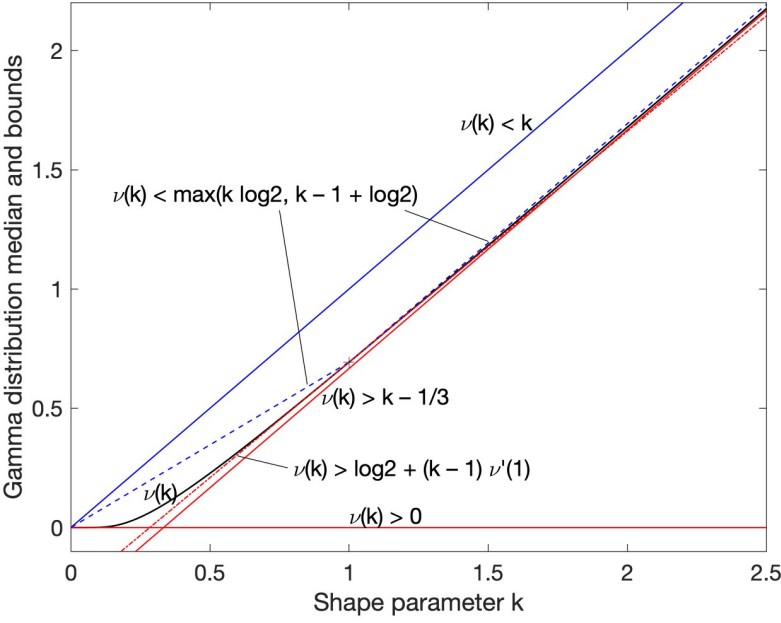

**Fig 1. Linear and piecewise-linear bounds.** The upper (blue) and lower (red) bounds $k - \frac{1}{3} < v(k) < k$ and $v(k) > 0$ (solid lines) are shown along with the true value (solid curve), the piecewise-linear bound that combines the recent linear bound for $k \geq 1$ [3] with a chord segment (dashed lines), and the linear lower bound that is tangent at $k = 1$ (dash-dot line). The region $k < 1$ is not very usefully bounded.

**Table 1. Bounds table.** Summary comparison of several upper and lower bounds. Gr&M refers to Groeneveld and Meeden 1977 [11], C&R refers to Chen and Rubin 1986 [4], B&P refers to Berg and Pedersen 2006 [6], and Ga&M refers to Gaunt and Merkle 2021 [3]. The * refers to bounds presented with informal proofs or derivations, and ** for bounds presented as conjectures without proof, in Lyon 2021 [1]; in the present paper they are treated as new theorems, with proofs.

| Upper bounds, with names and formulae | Domain | Tight at | Notes |
|---|---|---|---|
| $v(k) < U_0(k) = k$ | $k > 0$ | $k \to 0$ | Gr&M, C&R |
| $v(k) < U_1(k) = ke^{-1/(3k)}$ | $k > 0$ | $k \to \infty$ | B&P |
| $v(k) \leq U_2(k) = \log(2) + (k - 1)$ | $k \geq 1$ | $k = 1$ | Ga&M |
| $v(k) \leq U_3(k) = \log(2)k$ | $k \leq 1$ | $k = 1$ | Theorem U3* |
| $v(k) < U_4(k) = 2^{-1/k}\left(\frac{\Gamma(k+1)}{1-\frac{k}{k+1}U_1(k)}\right)^{1/k}$ | $0 < k \leq 1$ | $k \to 0$ | Theorem U4 |
| $v(k) < U_5(k) = 2^{-1/k}\left(\frac{\Gamma(k+1)}{1-\frac{k}{k+1}U_3(k)}\right)^{1/k}$ | $0 < k \leq 1$ | $k \to 0$ | Theorem U5 |
| $v(k) < U_6(k) = 2^{-1/k}(e^{-\gamma} + k)$ | $k > 0$ | $k \to 0$ | Theorem U6** |
| Lower bounds, with names and formulae | Domain | Tight at | Notes |
| $v(k) > L_0(k) = 0$ | $k > 0$ | $k \to 0$ | trivial lower bound |
| $v(k) > L_1(k) = k - \frac{1}{3}$ | $k > 0$ | $k \to \infty$ | Doodson 1917 [12], C&R |
| $v(k) > L_2(k) = 2^{-1/k}k$ | $k > 0$ | $k \to 0$ | B&P |
| $v(k) \geq L_3(k) = \log(2) + (k - 1)v'(1)$ | $k > 0$ | $k = 1$ | Theorem L3* |
| $v(k) > L_4(k) = 2^{-1/k}\Gamma(k + 1)^{1/k}$ | $k > 0$ | $k \to 0$ | Theorem L4* |
| $v(k) > L_5(k) = 2^{-1/k}e^{-\gamma}$ | $k > 0$ | $k \to 0$ | B&P asymptote, Theorem L5* |
| $v(k) > L_6(k) = 2^{-1/k}\left(\frac{\Gamma(k+1)}{1-\frac{1-e^{-k}}{k+1}L_3(k)}\right)^{1/k}$ | $k > 0$ | $k \to 0$ | Theorem L6 |
| $v(k) > L_7(k) = v_{Li} + k - k_i$ | $0 < k \leq k_i$ | — | Theorem L7 (for $v_{Li} < v(k_i)$) |
| $v(k) > L_8(k) = 2^{-1/k}(\log(2) - 1/3 + k)$ | $k > 0$ | $k \to \infty$ | Theorem L8** |

We prove the new theorems exhibited in Table 1, culminating in Theorems U6 and U8.

## Recent related work

In addition to the works mentioned above, there have been several more recent works on asymptotic properties and bounds for medians and other quantiles of gamma distributions and of the closely related Poisson and negative binomial (or Polya or Pascal) distributions, with a variety of interesting approaches [8–10].

# Theorems and proofs

## Chords and tangents

The convexity of the median (i.e., nonnegative second derivative) proved by Berg and Pedersen [7] implies that any tangent line is a lower bound, tight at the point of tangency, and that any chord, the straight line segment defined by two points of intersection, is an upper bound over the $k$ interval delimited by the points of intersection.

The point $k = 1$ with $v(1) = \log(2)$ is a point for which we have a known value, so is a good place to make a tangent lower bound. We can also use it for a chord with the other point of intersection at $k \to 0$ or $k \to \infty$. The Gaunt and Merkle upper bound [3] $U_2(k) = \log(2) + (k - 1)$ can be viewed as the limit of chords with a point of intersection at $k = 1$ and the other at $k \to \infty$ where the slope approaches 1. At the zero end is our $U_3(k)$, a rather trivial but apparently new observation.

**Theorem U3.** *The median of the standard gamma distribution is bounded above by the chord between $k = 0$ and $k = 1$:*

$$v(k) \leq \log(2)k \quad \text{for} \quad 0 < k \leq 1 \,.$$

*Proof*: The convexity of the median implies that a chord is an upper bound, between its points of intersection, tight at those points. The point $k = 1$, $v(1) = \log(2)$ (from the median of the exponential distribution, a well-known result and an easy computation), and the limiting point at $k = 0$, $v(0) = 0$, are the only places we have definite known expressions for the value of the median, so we can provide a formula for that chord. The straight line between $(0, 0)$ and $(1, \log(2))$ is the formula given.

**Theorem L3.** *The median of the standard gamma distribution is bounded below by the tangent line at $k = 1$:*

$$v(k) \geq \log(2) + (k - 1)v'(1) \quad \text{with} \quad v'(1) = \gamma - 2\text{Ei}(-\log(2)) - \log(\log(2)) \,.$$

*Proof*: The convexity of the median [7] implies that a tangent line is a lower bound, tight at the point of tangency. At $k = 1$ we know $v(1) = \log(2)$ and can compute the slope to form the equation for the tangent line, $\log(2) + (k - 1)v'(1)$.

The slope $v'(k)$ is not generally tractable, but is at the special point $k = 1$, where the CDF $P_k(x)$ (the lower incomplete gamma function) and PDF $p_k(x)$ are both exponential functions.

$$P_k(x) = \int_0^x p_k(t)dt = \int_0^x \frac{t^{k-1}}{\Gamma(k)} e^{-t} dt \,.$$

At the point where $P_k(x) = \frac{1}{2}$, where $x = v(k)$, the slope is:

$$v'(k) = \frac{dv}{dk} = -\frac{\partial P_k(x)}{\partial k} \bigg/ \frac{\partial P_k(x)}{\partial x} \,.$$

The derivative with respect to $x$ is easy,

$$\frac{\partial P_k(x)}{\partial x} = p_k(x) = \frac{x^{k-1}}{\Gamma(k)} e^{-x} \,,$$

except that we only have a closed-form relation between $x$ and $k$ at $k = 1$, where we know $x = v(1) = \log(2)$ and $p_k(x) = e^{-\log(2)} = \frac{1}{2}$, so the derivative is $\frac{1}{2}$ there. The derivative with respect to $k$ is messier:

$$\frac{\partial P_k(x)}{\partial k} = -\Gamma(k)^{-2} \frac{d\Gamma(k)}{dk} \int_0^x t^{k-1} e^{-t} dt + \Gamma(k)^{-1} \int_0^x \frac{dt^{k-1}}{dk} e^{-t} dt \,.$$

At $k = 1$, using $\Gamma(k) = 1$ and $d\Gamma(k)/dk = -\gamma$, this derivative evaluates to

$$\frac{\partial P_k(x)}{\partial k}\bigg|_{k=1} = \frac{\gamma}{2} + \int_0^x \log t \, e^{-t} dt$$

$$= \frac{\gamma}{2} + \text{Ei}(-\log(2)) - \gamma - \frac{1}{2}\log(\log(2)) \,,$$

where $\text{Ei}(-\log(2)) \approx -0.3786710$ is the exponential integral (integration and evaluation assisted by Wolfram Alpha and independently verified). Putting these results together we

get the slope of the median at 1, and hence the slope of the tangent-line lower bound there:

$$v'(k)|_{k=1} = \gamma - 2\text{Ei}(-\log(2)) + \log(\log(2)) \approx 0.9680448 \, .$$

## Using bounds for the exponential

To find new bounds for the median, we can bound the exponential in the integrand. Since $e^{-x}$ is a decreasing function of $x$ (derivative is $-e^{-x} < 0$), we can upper bound it for $x > 0$ by its starting value: $e^{-x} < 1$. Since it is convex (second derivative is $e^{-x} > 0$), we can lower bound it by a tangent line: $1 - x < e^{-x}$, and can upper bound it over the interval $0 < x < k$ by a chord: $e^{-x} < 1 - x(1 - e^{-k})/k$.

In the proofs below, the superscripts $a$, $b$, and $c$ are just names, not exponents.

**Theorem L4.** *The median of the standard gamma distribution is bounded below by*:

$$v(k) > L_4(k) = 2^{-1/k}\Gamma(k+1)^{1/k} \quad \text{for all} \quad k > 0 \, .$$

*Proof*: Use the constant upper bound to the exponential, $e^{-x} < 1$ for $x > 0$, in the CDF integrand, to notice this inequality:

$$\frac{1}{\Gamma(k)}\int_0^{v(k)} x^{k-1}dx > \frac{1}{\Gamma(k)}\int_0^{v(k)} e^{-x}x^{k-1}dx = \frac{1}{2} \, ,$$

which integrates to:

$$\frac{v(k)^k}{\Gamma(k+1)} > \frac{1}{2} \, .$$

Since the denominator and the exponent are positive, this expression can be decreased to achieve equality to one-half by substituting for $v(k)$ a positive function of sufficiently lower value, which we'll call $L_4(k)$:

$$\frac{L_4(k)^k}{\Gamma(k+1)} = \frac{1}{2} \quad \Rightarrow \quad L_4(k) < v(k) \quad \text{for all} \quad k > 0 \, .$$

Solving, we find the expression given: $L_4(k) = 2^{-1/k}\Gamma(k+1)^{1/k}$.

We choose to write the result factored this way, rather than a single fraction with exponent $1/k$ or $-1/k$, because the two parts emphasize the shape of the median function in two regions: $2^{-1/k}$, which is just a small fraction bigger than Berg and Pedersen's asymptote $L_5(k)$ [6] at low $k$, and $\Gamma(k+1)^{1/k}$, which is just a small offset below the Chen and Rubin straight-line bound $k - \frac{1}{3}$ [4] at high $k$. Most of our other new bounds keep the $2^{-1/k}$ factor, which characterizes the "hockey stick" shape at low $k$.

**Theorem L5.** *The median of the standard gamma distribution, and its lower bound $L_4(k)$, are bounded below by Berg and Pedersen's asymptote $L_5(k)$*:

$$v(k) > L_4(k) > L_5(k) = 2^{-1/k}e^{-\gamma} \, .$$

*Proof*: Comparing to Theorem L4, $L_5(k) < L_4(k)$ is implied if $e^{-\gamma} < \Gamma(k+1)^{1/k}$ for all $k > 0$. We prove this by showing that $\Gamma(k+1)^{1/k}$ monotonically increases from $e^{-\gamma}$, its limit at 0.

That $\lim_{k\to 0} \Gamma(k+1)^{1/k} = e^{-\gamma}$ follows from the Taylor series about 0 of $\Gamma(k+1)$, which is $1 - \gamma k + O(k^2)$. That $\Gamma(k+1)^{1/k}$ increases monotonically from there, even though $\Gamma(k+1)$ is decreasing, is proved by showing that its derivative is everywhere positive. Differentiating, in terms of the digamma function $\psi^{(0)}$, the logarithmic derivative of the gamma function, we

have:

$$\frac{d}{dk}\Gamma(k+1)^{1/k} = \frac{\Gamma(k+1)^{1/k}(k\psi^{(0)}(k+1) - \log(\Gamma(k+1)))}{k^2} \,.$$

The only factor here that is not obviously positive for $k > 0$ is $k\psi^{(0)}(k+1) - \log(\Gamma(k+1))$, which at $k = 0$ is equal to 0, and which has a surprisingly simple derivative:

$$\frac{d}{dk}\left(k\psi^{(0)}(k+1) - \log(\Gamma(k+1))\right) = k\psi^{(1)}(k+1) \,.$$

Here $\psi^{(1)}$ is the trigamma function, the derivative of the digamma function. This derivative is positive since the trigamma function, a special case of the Hurwitz zeta function, is positive for real arguments, because it has a series expansion with all positive terms:

$$\psi^{(1)}(z) = \sum_{n=0}^{\infty} \frac{1}{(z+n)^2} \,.$$

Since it starts at zero and has a positive derivative everywhere, the factor in question is positive for $k > 0$, so $\Gamma(k+1)^{1/k}$ is monotonically increasing.

**Theorem U4.** *The median of the standard gamma distribution is bounded above by this expression, which is asymptotic at low k to the lower bound $L_4(k)$:*

$$v(k) < U_4(k) = 2^{-1/k}\left(\frac{\Gamma(k+1)}{1 - \frac{k^2}{k+1}e^{-1/(3k)}}\right)^{1/k} \quad \text{when} \quad k \le 1 \,.$$

*Proof*: Use the tangent-line-at-0 lower bound to the exponential, $1 - x < e^{-x}$, in the integrand, to notice this inequality with easy integrals:

$$\frac{1}{\Gamma(k)}\int_0^{v(k)}(1-x)x^{k-1}dx < \frac{1}{\Gamma(k)}\int_0^{v(k)}e^{-x}x^{k-1}dx = \frac{1}{2}$$

$$\Rightarrow \quad \frac{v(k)^k}{\Gamma(k+1)} - \frac{v(k)^{k+1}k}{\Gamma(k+1)(k+1)} < \frac{1}{2} \,.$$

In this difference of terms, their exists a $U^a(k)$ such that the first term can be increased to achieve equality to one-half by substituting $U^a(k)$ for $v(k)$:

$$\frac{U^a(k)^k}{\Gamma(k+1)} - \frac{v(k)^{k+1}k}{\Gamma(k+1)(k+1)} = \frac{1}{2} \quad \Rightarrow \quad U^a(k) > v(k) \,.$$

Now, picking any known upper bound $U^b(k) \ge v(k)$, we have $U^a(k)^k U^b(k) > v(k)^{k+1}$, so we can write this inequality where the subtracted term has been increased:

$$\frac{U^a(k)^k}{\Gamma(k+1)} - \frac{U^a(k)^k U^b(k)k}{\Gamma(k+1)(k+1)} < \frac{1}{2}$$

$$\frac{U^a(k)^k}{\Gamma(k+1)}\left(1 - \frac{U^b(k)k}{k+1}\right) < \frac{1}{2} \,.$$

As long as both factors are positive, there exists a $U^c(k)$ such that we can increase the first factor by substituting $U^c(k)$ for $U^a(k)$ to achieve equality:

$$\frac{U^c(k)^k}{\Gamma(k+1)}\left(1 - \frac{U^b(k)k}{k+1}\right) = \frac{1}{2} \quad \Rightarrow \quad U^c(k) > U^a(k) > v(k)\,.$$

So we can solve for this new bound $U^c(k)$ in terms of the known bound $U^b(k)$:

$$v(k) < U^c(k) = 2^{-1/k}\left(\frac{\Gamma(k+1)}{1 - \frac{k}{k+1}U^b(k)}\right)^{1/k} \quad \text{as long as} \quad U^b(k)\frac{k}{k+1} < 1\,.$$

Using $U_1(k) = ke^{-1/(3k)}$ as the known bound $U^b$, and verifying the positivity constraint by noting that $U_1(k) < 1$ for $k \leq 1$ completes the proof.

By this method, we've taken an upper bound that's relatively loose at low $k$ and converted it to one that's asymptotically tight, approaching the lower bound $L_4(k)$, at low $k$. Let's do another like that.

**Theorem U5.** *The median of the standard gamma distribution is bounded above by*:

$$v(k) < U_5(k) = 2^{-1/k}\left(\frac{\Gamma(k+1)}{1 - \frac{k^2}{k+1}\log(2)}\right)^{1/k} \quad \text{when} \quad k \leq 1\,.$$

*Proof*: Same as Theorem U4, but use $U_3(k) = k\log(2)$ as the known upper bound $U^b(k)$. Note that $U_3(k)$ is a bound only for $k \leq 1$, and that the positivity constraint $U^c(k)\frac{k}{k+1} < 1$ holds through $k \leq 1$, since both factors $U^c(k)$ and $\frac{k}{k+1}$ are less than 1 in that domain.

**Theorem L6.** *The median of the standard gamma distribution is bounded below by*:

$$L_6(k) = 2^{-1/k}\left(\frac{\Gamma(k+1)}{1 - \frac{1-e^{-k}}{k+1}L_3(k)}\right)^{1/k} < v(k) \quad \text{for all} \quad k > 0\,.$$

*Proof*: Like Theorem U4, but with an upper, as opposed to lower, bound on the exponential, and changing all the directions of the inequalities. That is, use the chord from $x = 0$ to $x = k$ upper bound to the exponential, $e^{-x} < 1 - x\frac{1-e^{-k}}{k}$, in the integrand, to notice this inequality with easy integrals:

$$\frac{1}{\Gamma(k)}\int_0^{v(k)}\left(1 - x\frac{1-e^{-k}}{k}\right)x^{k-1}dx > \frac{1}{\Gamma(k)}\int_0^{v(k)}e^{-x}x^{k-1}dx = \frac{1}{2}$$

$$\Rightarrow \quad \frac{v(k)^k}{\Gamma(k+1)} - \frac{v(k)^{k+1}(1-e^{-k})}{\Gamma(k+1)(k+1)} > \frac{1}{2}\,.$$

In this difference of terms, the first term can be decreased to achieve equality to one-half by substituting for $v(k)$ a sufficiently smaller (but positive) $L^a(k)$:

$$\frac{L^a(k)^k}{\Gamma(k+1)} - \frac{v(k)^{k+1}(1-e^{-k})}{\Gamma(k+1)(k+1)} = \frac{1}{2} \quad \Rightarrow \quad L^a(k) < v(k)\,.$$

Now, picking any other lower bound $L^b(k) \le v(k)$, we have $L^a(k)^k L^b(k) < v(k)^{k+1}$ (even if $L^b(k) < 0$), so we can write this inequality where the subtracted term has been decreased:

$$\frac{L^a(k)^k}{\Gamma(k+1)} - \frac{U^a(k)^k L^b(k)(1 - e^{-k})}{\Gamma(k+1)(k+1)} > \frac{1}{2}$$

$$\Rightarrow \quad \frac{L^a(k)^k}{\Gamma(k+1)} \left( 1 - \frac{L^b(k)(1 - e^{-k})}{k+1} \right) > \frac{1}{2} \ .$$

As long as both factors are positive we can decrease the first factor by substituting a sufficiently smaller $L^c(k)$ to achieve equality:

$$\frac{L^c(k)^k}{\Gamma(k+1)} \left( 1 - \frac{L^b(k)(1 - e^{-k})}{k+1} \right) = \frac{1}{2} \quad \Rightarrow \quad L^c(k) < L^a(k) < v(k) \ .$$

So we can solve for a new lower bound $L^c(k)$ in terms of a known lower bound $L^b(k)$:

$$v(k) > L^c(k) = 2^{-1/k} \left( \frac{\Gamma(k+1)}{1 - \frac{(1 - e^{-k})}{k+1} L^b(k)} \right)^{1/k} \quad \text{as long as} \quad \frac{(1 - e^{-k})}{k+1} L^b(k) < 1 \ .$$

The constraint is met for all positive $k$, with any lower bound $L^b(k)$, since $L^b(k) < v(k) < k$ $\Rightarrow L^b(k)/(k+1) < 1$.

Using $L_3(k) = \log(2) + (k-1)v'(1)$, the tangent at 1, as the known bound $L^b$ completes the proof.

$$v(k) > L_6(k) = 2^{-1/k} \left( \frac{\Gamma(k+1)}{1 - \frac{k^2}{k+1}(\log(2) + (k-1)v'(1))} \right)^{1/k} \quad \text{for all positive } k \ .$$

**Corollary 1**

$$v(k) > L_{61}(k) = 2^{-1/k} \left( \frac{\Gamma(k+1)}{1 - \frac{k^2}{k+1}(\log(2) + k - 1)} \right)^{1/k} \quad \text{for} \quad k \le 1 \ .$$

*Proof*: Use $L^b(k) = \log(2) + k - 1 < L_3(k)$, a lower bound to the tangent at $k = 1$ for $k \le 1$, since the slope 1 exceeds the tangent-line slope $v'(1)$.

We could obviously write some more corollaries, replacing negative lower bounds by zero, i.e. using $\max(0, L_3(k))$ or $\max(0, \log(2) + k - 1)$ as $L^b(k)$. Where the negative values are clipped to zero, the bound will turn to follow the tighter $L_4(k)$.

## Bounds in a box

Our theorems U6 and L8 follow if we can prove that

$$\log(2) - \frac{1}{3} < A(k) < e^{-\gamma} \quad \text{where} \quad A(k) = 2^{1/k} v(k) - k \ .$$

These bounds $A_L = \log(2) - \frac{1}{3}$ and $A_U = e^{-\gamma}$ define the bottom and top edges of a "box" that we need $A(k)$ to be constrained to, and we can visualize that by mapping other bounds through the same function, and plot them, and show that at least one is inside the box for any and all $k$. See Fig 2.

The function that maps the median and its bounds is $f(k, x(k)) = 2^{1/k} x(k) - k$, where $x(k)$ is any real-valued function of positive $k$. We map bounds into the same space as $A(k)$ with

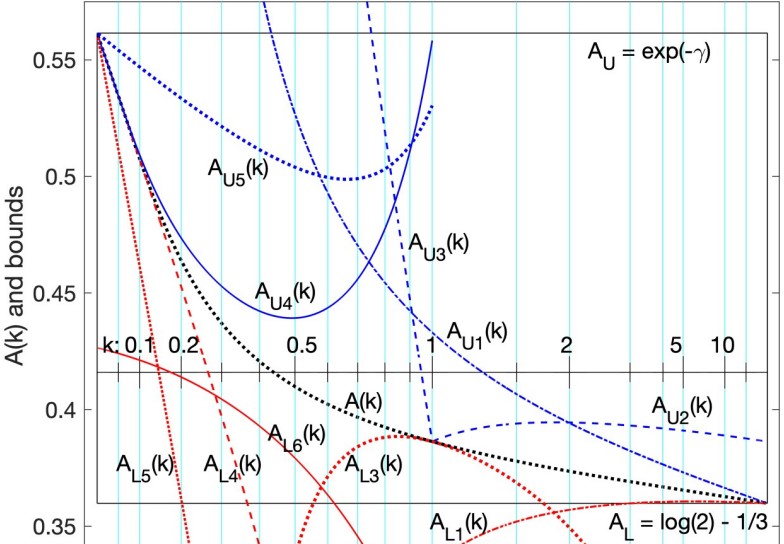

**Fig 2. Bounds in a box.** Upper (blue) and lower bounds (red) mapped for comparison to $A(k)$ (black dotted). Bounds $U_6(k)$ and $L_8(k)$ define the top and bottom of the box via $A_{U6}(k) = A_U = e^{-\gamma}$ and $A_{L8}(k) = A_L = \log(2) - 1/3$, while the left and right are defined by the limits of $\arctan(k)$ for $0 < k < \infty$. To prove that the top and bottom are bounds of $A(k)$, our approach is to find other upper and lower bounds "inside the box" over domains covering all $k > 0$. In this figure, we have no lower bound in the box around $1.7 < k < 3.0$.

this, and identify them with subscripts. In particular, consider these upper bounds (plotted in Fig 2):

$$A_{U1}(k) = 2^{1/k} U_1(k) - k = 2^{1/k} k e^{-1/(3k)} - k,$$

$$A_{U2}(k) = 2^{1/k} U_2(k) - k = 2^{1/k}(\log(2) + (k-1)) - k,$$

$$A_{U3}(k) = 2^{1/k} U_3(k) - k = 2^{1/k}(k\log(2)) - k,$$

$$A_{U4}(k) = 2^{1/k} U_4(k) - k = \left( \frac{\Gamma(k+1)}{1 - \frac{k}{k+1} U_1(k)} \right)^{1/k} - k,$$

$$A_{U5}(k) = 2^{1/k} U_5(k) - k = \left( \frac{\Gamma(k+1)}{1 - \frac{k}{k+1} U_3(k)} \right)^{1/k} - k.$$

It's easy to see, graphically, that $A_{U1}(k)$ and $A_{U2}(k)$ are "in the box" for $k \geq 1$, and that $A_{U4}(k)$ and $A_{U5}(k)$ are "in the box" for $k \leq 1$. Whether it's easy to prove is another matter. At least we're working with well-defined expressions and functions with known properties, not with the implicitly defined $v(k)$ itself.

We can do the same for some lower bounds, and see where they end up relative to the box:

$$A_{L1}(k) = 2^{1/k}L_1(k) - k = 2^{1/k}\left(k - \frac{1}{3}\right) - k\,,$$

$$A_{L3}(k) = 2^{1/k}L_3(k) - k = 2^{1/k}(\log(2) + (k-1)v'(1)) - k\,,$$

$$A_{L4}(k) = 2^{1/k}L_4(k) - k = \Gamma(k+1)^{1/k} - k\,,$$

$$A_{L6}(k) = 2^{1/k}L_6(k) - k = \left(\frac{\Gamma(k+1)}{1 - \frac{1-e^{-k}}{k+1}L_3(k)}\right)^{1/k} - k\,.$$

The lower bounds $L_1(k)$, $L_3(k)$, and $L_4(k)$ are well inside the box (greater than $A_L = \log(2) - \frac{1}{3}$) for $k$ near enough to $\infty$, 1, and 0, respectively, but they leave significant holes between them, where they don't constrain $A(k)$ against sagging out of the bottom of the box. That's why we resort to the additional complexity of $L_6(k)$ and $L_7(k)$, to fill in the holes.

## Line bounds from point bounds

It is straightforward to find or verify bounds for $v(k)$ at a finite set of discrete values of $k$, using numerical evaluation of the CDF integral via a rapidly converging series based on the Taylor series for the exponential function. For Theorem L7, we prove slope-1 line lower bounds based on point bounds at eight $k$ points, for $k$ values less than the point $k$ (for higher values of $k$, the line will cross to be greater than $v(k)$. We use these line bounds to supplement our other lower bounds, in service of proving Theorem L8.

**Theorem L7.** *The median of the standard gamma distribution is bounded below by lines of slope 1 below point lower bounds*:

$$v(k) > L_7(k) = v_{Li} + k - k_i \quad \text{for} \quad k \le k_i\,,$$

*where the values $k_i$ and $v_{Li}$ represent eight point lower bounds $v_{Li} < v(k_i)$ per this array of values*:

| $i$ | $k_i$ | $v_{Li}$ |
|-----|-------|----------|
| 1 | 0.40 | 0.145 |
| 2 | 0.44 | 0.177 |
| 3 | 0.50 | 0.227 |
| 4 | 0.60 | 0.315 |
| 5 | 0.75 | 0.454 |
| 6 | 1.00 | 0.693 |
| 7 | 1.50 | 1.182 |
| 8 | 3.50 | 3.172 . |

*Proof*: Since the slope of $v(k)$ is everywhere less than 1, these lines of slope 1 below point lower bounds are below the tangent line lower bounds at those points. To demonstrate that each $v_{Li}$ is indeed a lower bound at each of the $k_i$ points, we use a transparent technique not relying on anybody's implementation of incomplete gamma functions. The point lower bounds are verified to lead to percentiles (100 times the CDF estimate) below 50%, by more

than the corresponding estimation error bounds. In each case, we have chosen the $v_{Li}$ to be the largest multiple of 0.001 that can be verified to be a lower bound.

These calculations are shown in the S1 Appendix, with intermediate steps printed out from the algorithm given there. The formulae used for the CDF include approximations for the incomplete gamma function integral. The incomplete gamma function integral in the CDF can be written as a convergent series, using the Taylor series of the exponential function:

$$\int_0^v x^{k-1} e^{-x} dx = \int_0^v x^{k-1} \sum_{n=0}^{\infty} \frac{(-x)^n}{n!} dx = \sum_{n=0}^{\infty} \frac{(-1)^n}{n!} \int_0^v x^{n+k-1} dx = \sum_{n=0}^{\infty} \frac{(-1)^n v^{n+k}}{n!(n+k)} .$$

Since the magnitudes of the terms are decreasing by half or better after enough terms that $n > 2v$, the error in truncating the series is bounded by the magnitude of the last term added (and due to the alternation, the error is actually quite a bit lower than that). So to verify a point lower bound $v < v(k)$, we accumulate terms until that condition is met and the sum plus the bound on the residue, divided by a lower bound of $\Gamma(k)$, is less than 0.5.

We lower-bound $\Gamma(k)$ at these points by truncating values from a standard math package to 5 decimal digits. We also computed these bounds from scratch, using Euler's product formula with 80 or more factors, and using upper and lower bounds on the truncated tail residual factor computed by using integrals to bound the sum of logarithms of the factors. That analysis is too long and complicated to include, but we list the values we used so that others can independently verify them.

See the S1 Appendix for the numerical verifications of these values by this method.

The reason for the particular set of $k_i$ selected for this theorem will become clear later (or may be obvious from looking at Fig 3).

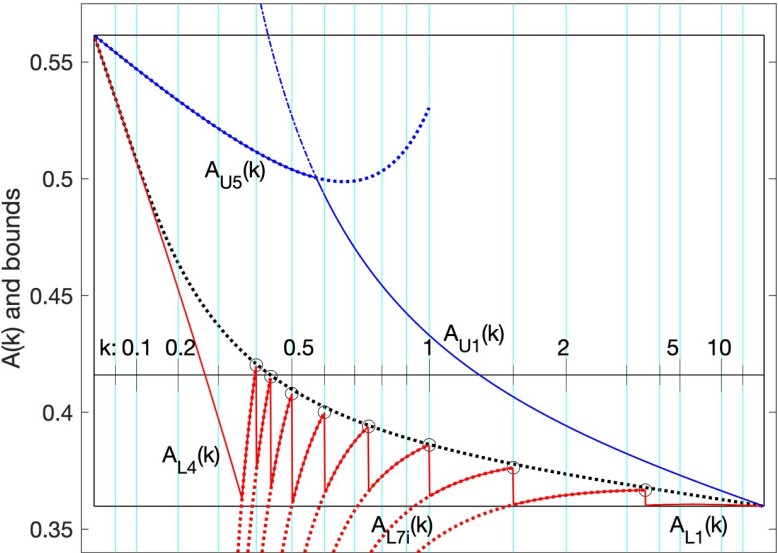

**Fig 3. More bounds in the box.** Here we show some approximate-tangent lower bounds from Theorem L7 mapped to the box, and remove some of the others. The points $\{k_i, v_{Li}\}$ of Theorem 7 are circled. The solid blue upper bound is in the box per Lemma 1 and Lemma 2. The solid red lower bound is in the box per Lemmas 3, 4, and 5. These bounds being in the box will prove our main theorems U6 and L8.

## Piecing the bounds together

This is where it gets complicated. We have to find regions where various of the previous bounds are tighter than the main upper and lower bounds that we set out to prove. To this end we will prove the lemmas in Table 2.

**Lemma 1.**

$$A_{U1}(k) = 2^{1/k}ke^{-1/(3k)} - k < e^{-\gamma} \quad \text{for} \quad k \geq 0.5.$$

*Proof*:

$$A_{U1}(k) = 2^{1/k}ke^{-1/(3k)} - k = ke^{(\log(2)-1/3)/k} - k$$

$$\Rightarrow \quad \frac{d}{dk}A_{U1}(k) = e^{(\log(2)-1/3)/k}\frac{k - \log(2) + 1/3}{k} - 1.$$

The exponential can be upperbounded using $e^x < 1/(1-x)$ for $x < 1$:

$$\frac{d}{dk}A_{U1}(k) < \frac{1}{1 - (\log(2) - 1/3)/k}\frac{k - \log(2) + 1/3}{k} - 1 \quad \text{for} \quad k > \log(2) - 1/3$$

$$\Rightarrow \quad \frac{d}{dk}A_{U1}(k) < 0 \quad \text{for} \quad k > 0.360 > \log(2) - 1/3.$$

Concluding that the slope of $A_{U1}(k)$ is negative for $k > 0.360$, and evaluating $A_{U1}(0.5) < 0.527$ and $e^{-\gamma} > 0.561$, we can conclude that $A_{U1}(k) < e^{-\gamma}$ for $k \geq 0.5$.

**Lemma 2.**

$$A_{U5}(k) = 2^{1/k}U_5(k) - k < e^{-\gamma} \quad \text{for} \quad k \leq 1.$$

*Proof*:

$$A_{U5}(k) = 2^{1/k}U_5(k) - k = \left(\frac{\Gamma(k+1)}{1 - \frac{k^2}{k+1}\log(2)}\right)^{1/k} - k.$$

The low-$k$ limit is again $e^{-\gamma}$; the slope is initially negative, but the $A_{U5}(k)$ eventually crosses above the limit and diverges to infinity where the denominator expression goes to 0. To show that it doesn't cross above the limit until somewhere after $k = 0.5$, we'll use several steps of bounding of functions and derivatives, starting by invoking an upper bound on the gamma

**Table 2. Lemmas table.** Lemmas to prove in support of Theorems U6 and L8.

| Upper bounds in box | Domain | Lemma |
|---|---|---|
| $A_{U1}(k) < e^{-\gamma}$ | $k \geq 0.5$ | Lemma 1 |
| $A_{U5}(k) < e^{-\gamma}$ | $k \leq 1$ | Lemma 2 |
| Lower bounds in box | Domain | |
| $A_{L1}(k) > \log(2) - 1/3$ | $k \geq 3.1$ | Lemma 3 |
| $A_{L4}(k) > \log(2) - 1/3$ | $0 < k \leq 0.36$ | Lemma 4 |
| $A_{L7i}(k) > \log(2) - 1/3$ | $k_{i-1} \leq k \leq k_i, k_0 = 0.36$ | Lemma 5 |

function, Theorem 1.4 of Batir [13]:

$$\Gamma(k+1) \le (e^{-\gamma})^{(-e^{-\gamma})} e^{-k} (k+e^{-\gamma})^{(k+e^{-\gamma})} \,.$$

The condition $A_{U5}(k) \le e^{-\gamma}$ is satisfied if substituting this upper bound satisfies the constraint:

$$\left( \frac{(e^{-\gamma})^{(-e^{-\gamma})} e^{-k} (k+e^{-\gamma})^{(k+e^{-\gamma})}}{1 - \frac{k^2}{k+1}\log(2)} \right)^{1/k} - k \le e^{-\gamma}$$

or

$$\frac{(e^{-\gamma})^{(-e^{-\gamma})} e^{-k} (k+e^{-\gamma})^{(k+e^{-\gamma})}}{1 - \frac{k^2}{k+1}\log(2)} \le (k+e^{-\gamma})^k \,.$$

(since the denominator is positive in the domain we care about, $0 < k < 1$). Dividing by the positive right-hand side gives the equivalent condition

$$\frac{(e^{-\gamma})^{(-e^{-\gamma})} e^{-k} (k+e^{-\gamma})^{e^{-\gamma}}}{1 - \frac{k^2}{k+1}\log(2)} \le 1 \,,$$

which simplifies to

$$\frac{e^{-k}(ke^{\gamma}+1)^{e^{-\gamma}}}{1 - \frac{k^2}{k+1}\log(2)} \le 1$$

or

$$g(k) = (k+1)\left(e^{-k}(ke^{\gamma}+1)^{e^{-\gamma}}\right) - (k+1) + k^2\log(2) \le 0 \,.$$

This function has a Taylor series at 0 starting with a negative $k^2$ term and a positive $k^3$ term, so it is negative in some region of low enough $k$, as needed. By showing that the third derivative stays positive in $0 < k < 1$, we can conclude that it crosses its starting value not more than once, and we can bound where that happens with a few evaluations. So first we need the derivatives; with help from WolframAlpha:

$$g'(k) = e^{-k}(1 - e^{\gamma}k^2)(e^{\gamma}k+1)^{e^{-\gamma}-1} + 2k\log(2) - 1 \,,$$

$$g''(k) = \frac{e^{-k+e^{-\gamma}\gamma+\gamma}(e^{\gamma}k^3 - e^{\gamma}k^2 - 3k - 1)(k+e^{-\gamma})^{e^{-\gamma}}}{(e^{\gamma}k+1)^2} + 2\log(2) \,,$$

$$g'''(k) = e^{\gamma-k}(e^{\gamma}k+1)^{e^{-\gamma}-3}(2e^{\gamma}(3k^2+k+1) - e^{2\gamma}(k-2)k^3 - 3) \,.$$

To show $g'''(k) > 0$ over the domain of interest (say $0 < k < 0.5$), we can throw away the leading obviously-positive factors, and the condition becomes $2e^{\gamma}(3k^2+k+1) - e^{2\gamma}(k-2)k^3 - 3 > 0$, a simple fourth-degree polynomial-in-$k$ condition:

$$-e^{2\gamma}k^4 + 2e^{2\gamma}k^3 + 6e^{\gamma}k^2 + 2e^{\gamma}k + (2e^{\gamma} - 3) > 0 \,.$$

All coefficients except the highest-order one are positive. Without that fourth-order term, this polynomial would be positive for all $k \ge 0$, with one real root below zero; with that term, it has a positive real root, below which it is positive, including about 0.562 at $k = 0$ and about 17.98 at $k = 1$, so it's positive for at least $0 \le k \le 1$. This is all we need; let's unwind.

The positive $g'''(k)$ means that the curvature, $g''(k)$, is increasing between $k = 0$ and $k = 1$. The slope $g'(k)$ starts out negative at $k = 0$, keeping $g(k) < 0$ until after the slope eventually increases enough due to the eventually positive curvature. After the $g(k)$ becomes positive, it will stay that way until some time after $g'''(k)$ turns negative, making $g''(k)$ turn negative, etc., and this is well above the region of interest. Evaluating a few points, $g(0.5) = -0.0258$, $g(1) = -0.00025$, $g(1.01) = 0.0018$. So the original condition holds, $A_{U5}(k)$ is "in the box", through $k = 1$, the end of its domain of validity.

**Lemma 3.**

$$A_{L1}(k) = 2^{1/k}L_1(k) - k > \log(2) - 1/3 \quad \text{for} \quad k \geq 3.1.$$

*Proof*: $L_1(k)$ maps to $A_{L1}(k) = 2^{1/k}(k - 1/3) - k$. The Laurent series at infinity of this function is easily found (with the help of Wolfram Alpha) to be:

$$A_{L1}(k) = 2^{1/k}(k - 1/3) - k = \sum_{n \geq 0} \frac{\log^n(2)(\log(2) - 1/3 - n/3)}{k^n \ (n+1)!}$$

$$= \log(2) - 1/3 + \log(2)(\log(2)/2 - 1/3)k^{-1} + O(k^{-2}),$$

where the coefficient of $k^{-1}$ is positive and all of the coefficients in the $O(k^{-2})$ term are negative (where $n > 3\log(2) - 1$). Therefore, for sufficiently large $k$, $A_{L1}(k)$ exceeds $L_U = \log(2) - 1/3$, and for smaller $k$, the value can only cross below $\log(2) - 1/3 \approx 0.35981$ once due to all the higher-order coefficients being negative. Evaluating at $k = 3$ we find $A_{L1}(k) \approx 0.35979$, outside the box, and evaluating at $k = 3.1$ we find $A_{L1}(k) \approx 0.35990$, inside the box. Thus we can conclude that $A_{L1}(k)$ is inside the box for $k \geq 3.1$.

**Lemma 4.**

$$A_{L4}(k) = 2^{1/k}L_4(k) - k > \log(2) - 1/3 \quad \text{for} \quad 0 < k \leq 0.36.$$

*Proof*: $L_4(k)$ maps to $A_{L4}(k) = \Gamma(k+1)^{1/k} - k$, which approaches $e^\gamma$ (at the top of the box) as $k \to 0$, and exits the box somewhere in $0.36 < k < 0.37$ since $A_{L4}(0.36) \approx 0.3638 > 0.3598$ and $A_{L4}(0.37) \approx 0.3583 < 0.3598$. We just need to show it can only exit the box once in $0 < k < 0.37$ to conclude $A_{L4}(k) > \log(2) - 1/3$ for all $0 < k < 0.36$. $\Gamma(k+1)^{1/k}$ has a positive derivative, as we showed in the proof of Theorem L5. Now, if we can prove that derivative is less than 1, then $A_{L4}(k)$ is monotonically decreasing, so it can only go out of the box once.

$$\frac{d}{dk}\Gamma(k+1)^{1/k} = \Gamma(k+1)^{1/k}\frac{k\psi^{(0)}(k+1) - \log(\Gamma(k+1))}{k^2} < 1 \quad \text{in} \quad 0 < k < 1$$

because both factors are less than 1; $\Gamma(x) < 1$ in $1 < x < 2$ (a well-known property of the gamma function); to show $(k\psi^{(0)}(k+1) - \log(\Gamma(k+1))/k^2 < 1$, we need to do a bit more work. First, define the function $h(k)$ that this factor is the derivative of:

$$h(k) = \frac{\log(\Gamma(k+1))}{k} \quad \Rightarrow \quad \frac{d}{dk}h(x) = h'(k) = \left(k\psi^{(0)}(k+1) - \log(\Gamma(k+1))\right)/k^2.$$

Then rewrite the log of the gamma function in terms of this identity derived again from the Weierstrauss product formula and the zeta function:

$$\log(\Gamma(z+1)) = -\gamma z + \int_0^\infty \frac{e^{-zt} - 1 + zt}{t(e^t - 1)}dt$$

so that

$$h(k) = \frac{\log(\Gamma(k+1))}{k} = -\gamma + \int_0^\infty \frac{(e^{-kt}-1)/kt+1}{(e^t-1)}\, dt\ .$$

The $1/(e^t - 1)$ in the denominator of the integrand makes this integral absolutely convergent, so we can differentiate twice, getting:

$$h'(k) = \int_0^\infty \frac{-(kt+1)e^{-kt}+1}{k^2 t(e^t-1)}\, dt\ ,$$

$$h''(k) = \int_0^\infty \frac{[(kt+1)^2+1]e^{-kt}-2}{k^3 t(e^t-1)}\, dt\ ,$$

in which the integrand has a positive denominator and a negative numerator, since for any $k$ we can define $y = kt$ and write the numerator as $[(y+1)^2+1]e^{-y} - 2$, which starts at 0 and has derivative $-y^2 e^{-y} < 0$ for all $y > 0$. Since $h''(k) < 0$, $h'(k)$ only decreases from its starting value, which is less than 1, this completes the proof that $A_{L4}(k)$ decreases monotonically, so it is "in the box" for $0 < k \le 0.36$.

**Lemma 5.**

$$A_{L7i}(k) = 2^{1/k} L_{7i}(k) - k > \log(2) - 1/3 \quad \text{for} \quad k_{i-1} \le k \le k_i,\ k_0 = 0.36\ .$$

*Proof*: The lower bounds $L_{7i}(k) = v_{Li} + k - k_i$, lines of slope 1 that are above $L_1(k)$, are of the form $k - a$ for $a = v_{Li} - k_i$, and map under $f$ to a function with series similar to the one we saw for Lemma 3 where $a = 1/3$, but with $a$ values in the range $v_{L1} - k_1 = 0.255 \le a \le v_{L8} - k_8 = 0.328$.

$$A_{L7} = 2^{1/k}(k-a) - k = \sum_{n \ge 0} \frac{\log^n(2)(\log(2) - a - an)}{k^n(n+1)!}\ .$$

In Lemma 3 we saw that the coefficient of $k^{-1}$ was positive, as it is here, since $a < \log(2)/2 \approx 0.346$, and that all the higher-order coefficients are negative, since for $n \ge 2$, $(1+n)a > \log(2)$ for $a > \log(2)/3 \approx 0.231$. Thus, as with $A_{L1}(k)$ in Lemma 3, lines of slope 1 are not monotonic when mapped into the box, but are in the box at some $k$ and only leave the bottom of the box once if $a \le 1/3$; that is, if $L_{7i}(k) > L_1(k)$. Therefore, to verify that they are in the box over the domains specified, all that is needed is to evaluate them at both ends of their respective domains, as in this array:

| $i$ | $a$ | $k_i$ | $v_i = L_i(k_i)$ | $A_{Li}(k_i)$ | $k_{i-1}$ | $L_i(k_{i-1})$ | $A_{Li}(k_{i-1})$ |
|---|---|---|---|---|---|---|---|
| 1 | 0.255 | 0.40 | 0.145 | 0.4202 | 0.36 | 0.105 | 0.3601 |
| 2 | 0.263 | 0.44 | 0.177 | 0.4153 | 0.40 | 0.137 | 0.3750 |
| 3 | 0.273 | 0.50 | 0.227 | 0.4080 | 0.44 | 0.167 | 0.3670 |
| 4 | 0.285 | 0.60 | 0.315 | 0.4001 | 0.50 | 0.215 | 0.3600 |
| 5 | 0.296 | 0.75 | 0.454 | 0.3940 | 0.60 | 0.304 | 0.3651 |
| 6 | 0.307 | 1.00 | 0.693 | 0.3860 | 0.75 | 0.443 | 0.3663 |
| 7 | 0.318 | 1.50 | 1.182 | 0.3763 | 1.00 | 0.682 | 0.3640 |
| 8 | 0.328 | 3.50 | 3.172 | 0.3667 | 1.50 | 1.172 | 0.3604 |

The $A_{Li}(k_i)$ and $A_{Li}(k_{i-1})$ columns in this array being greater than $\log(2) - 1/3 \approx 0.3598$ proves that the lower bounds are "in the box" over the respective domains $k_{i-1} \le k \le k_i$.

**Theorem U6.** *The median of the standard gamma distribution is bounded above by $L_6(k)$:*

$$v(k) < U_6(k) = 2^{-1/k}(e^{-\gamma} + k) \quad \text{for all} \quad k > 0.$$

*Proof*: Relative to the top of the box $A_{U6}(k) = e^{-\gamma} \approx 0.5615$, from Lemma 1 and Lemma 2, $A_{U1}(k) < e^{-\gamma}$ for $k \geq 0.5$ and $A_{U5}(k) < e^{-\gamma}$ for $k \leq 1$, which imply $U_6(k) > U_1(k) > v(k)$ and $U_6(k) > U_5(k) > v(k)$ over the respective same domains of $k$, so we conclude $U_6(k) > v(k)$ for all $k > 0$.

It is left as an exercise for the reader to prove using $U_2(k)$ and/or $U_4(k)$, leading to $U_6(k) > U_2(k) \geq v(k)$ for $k \geq 1$ and/or $U_6(k) > U_4(k) > v(k)$ for $0 < k \leq 1$; or some other combination of bounds in the box.

**Theorem L8.** *The median of the standard gamma distribution is bounded below by $L_8(k)$:*

$$v(k) > L_8(k) = 2^{-1/k}(\log(2) - 1/3 + k) \quad \text{for all} \quad k > 0.$$

*Proof*: Relative to the bottom of the box $A_{L8}(k) = \log(2) - 1/3 \approx 0.3598$, Lemmas 3, 4, and 5 show other bounds "in the box" over regions covering all $k > 0$: $A_{L4}(k) > \log(2) - 1/3$ for $0 < k \leq 0.36$, eight versions of $A_{L7i}(k) > \log(2) - 1/3$ for eight adjacent domains spanning $0.36 \leq k \leq 3.5$, and $A_{L1}(k) > \log(2) - 1/3$ for $k > 3.1$, which imply $L_8(k) < L_4(k) < v(k)$, $L_8(k) < L_{7i}(k) < v(k)$, and $L_8(k) < L_1(k) < v(k)$ over the respective same $k$ domains, so we conclude $L_8(k) < v(k)$ for all $k > 0$.

It is left as an exercise for the reader to prove via some other combination of lower bounds in the box. The Theorem L6, or its simpler corollary, provides a bound in the box until L3 takes over. The Theorem L7 bound with just the one segment at $i = 8$ can fill in between L3 and L1.

**Better bounds.** From the previous figures it is clear that there is room for smooth functions bounding $A(k)$ above and below that are tighter (in some regions, especially in the middle) than the bounds we constructed to prove theorems U6 and L8. Lyon [1] explored rational-function and arctan interpolators to approximate or bound $A(k)$, writing $A(k)$ as a monotonic interpolation between its upper and lower limits via a function $g(k)$:

$$A(k) = g(k)A_L + (1 - g(k))A_U.$$

The simplest rational-function interpolator used was

$$\tilde{g}_1(k) = \frac{k}{b_0 + k}.$$

Based on slopes at 0 and infinity, Lyon showed that the $b_0$ value for a lower bound could not exceed $b_L = \left(\frac{8}{405} + e^{-\gamma}\log 2 - \frac{\log^2 2}{2}\right) / \left(e^{-\gamma} - \log 2 + \frac{1}{3}\right) - \log 2 \approx 0.143472$ and the value for an upper bound could not be less than $b_U = \left(e^{-\gamma} - \log 2 + \frac{1}{3}\right) / \left(1 - \frac{e^{-\gamma}\pi^2}{12}\right) \approx 0.374654$, but that these values lead to actual lower and upper bounds, respectively, has not been proved.

See Fig 4 for the relation between these interpolated approximations to $A(k)$ and the actual (numerical) $A(k)$. Conjectured bounds from arctan interpolators are also shown, using this formula to approximate or bound $g(k)$:

$$\tilde{g}_a(k) = \frac{2}{\pi}\tan^{-1}\frac{k}{b}.$$

where for an upper bound the $b$ can not be less than $b_U = (24/\pi)\left(e^{-\gamma} - \log 2 + \frac{1}{3}\right) / (12 - e^{-\gamma}\pi^2) \approx 0.238512$, and for a lower bound not more

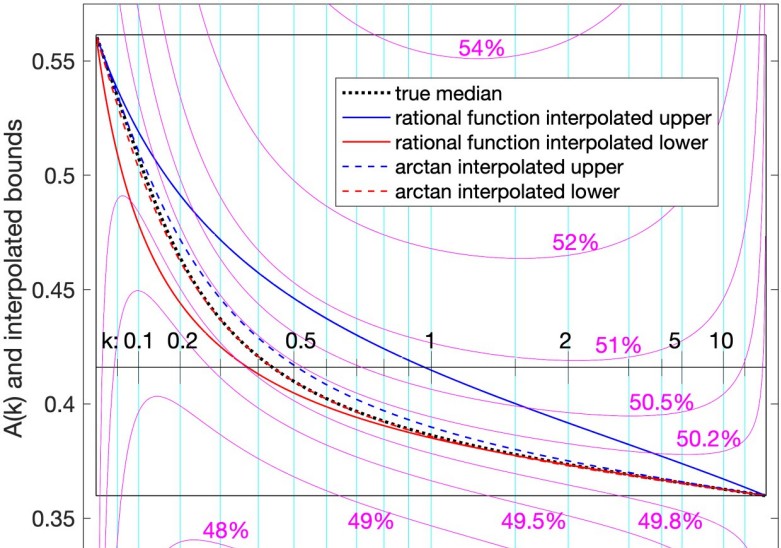

**Fig 4. Tighter bounds.** Conjectured better upper (blue) and lower (red) bounds mapped for comparison to $A(k)$ (black dotted). Solid curves are from first-order rational-function interpolators; dashed curves are from arctan interpolators. For a sense of how close these bounds come to the median (50th percentile), curves of selected nearby percentiles are included (thin black curves, computed with Matlab's `gammaincinv` function).

than $b_L \approx 0.205282$ (no closed form is known for this limit). The plots also show that the CDF, near 0.5, is surprisingly insensitive to the value of $A(k)$ near both $k \to 0$ and $k \to \infty$.

Again, that these interpolated functions produce actual bounds has not been proved. Presumably the method of line bounds from point bounds could be extended to construct proofs by finding tighter bounds numerically, though it might take thousands of points. These conjectures are left for others to consider.

## Conclusions

We give upper and lower bounds for the median of the gamma distribution that are tighter at low $k$ than previously known bounds, and are proved valid over all $k > 0$: $U_6(k)$ and $L_8(k)$. They are simple and in closed form. Though their validity seemed obvious from the numerical and asymptotic methods by which they were discovered, they were originally presented as conjectures [1] because they were not easy to prove analytically; the proofs here follow the outline of the proof the "hard way" as proposed there.

In summary, the median $v(k)$ of the standard gamma distribution satisfies

$$\log(2) - \frac{1}{3} < 2^{1/k} v(k) - k < e^{-\gamma} \quad \text{for all} \quad k > 0$$

or

$$2^{-1/k}\left(\log(2) - \frac{1}{3} + k\right) < v(k) < 2^{-1/k}(e^{-\gamma} + k).$$

This formulation for the median

$$v(k) = 2^{-1/k}(A(k) + k)$$

defines the function $A(k) = 2^{1/k}v(k) - k$ with the remaining conjecture that $A(k)$ is monotonically decreasing between the limits $\lim_{k\to 0}A(k) = e^{-\gamma}$ and $\lim_{k\to\infty}A(k) = \log(2) - 1/3$.

As we showed before [1], monotonic approximations to $A(k)$ that interpolate between these limits can make excellent approximations to the median, in closed form, with controlled good properties such as being exact at $k = 1$.

## Supporting information

**S1 Appendix.**
(PDF)

## Acknowledgments

Help and ideas from discussions with my mathematically talented colleagues at Google are gratefully acknowledged: Pascal Getreuer, Srinivas Vasudevan, Dan Piponi, Michael Keselman, Yuan Li, Thomas Fischbacher, Daniel Parry, Lizao Li, Fred Akalin, and John Vogler.

## Author Contributions

**Conceptualization:** Richard F. Lyon.

**Investigation:** Richard F. Lyon.

**Writing – original draft:** Richard F. Lyon.

**Writing – review & editing:** Richard F. Lyon.

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
