## [Decision Letter · Decision Letter 0]

22 May 2023

PONE-D-22-34782

Tight bounds for the median of a gamma distribution

PLOS ONE

Dear Dr. Lyon,

Thank you for submitting your manuscript to PLOS ONE. After careful consideration, we feel that it has merit but does not fully meet PLOS ONE’s publication criteria as it currently stands. Therefore, we invite you to submit a revised version of the manuscript that addresses the points raised during the review process.

You will see that the two reviewers are advising that you revise your manuscript. Note that it is a minor revision so please consider making the suggested changes.

A rebuttal letter that responds to each point raised by the academic editor and reviewer(s). You should upload this letter as a separate file labeled 'Response to Reviewers'.A marked-up copy of your manuscript that highlights changes made to the original version. You should upload this as a separate file labeled 'Revised Manuscript with Track Changes'.An unmarked version of your revised paper without tracked changes. You should upload this as a separate file labeled 'Manuscript'If applicable, we recommend that you deposit your laboratory protocols in protocols.io to enhance the reproducibility of your results. Protocols.io assigns your protocol its own identifier (DOI) so that it can be cited independently in the future. For instructions see: https://journals.plos.org/plosone/s/submission-guidelines#loc-laboratory-protocols. Additionally, PLOS ONE offers an option for publishing peer-reviewed Lab Protocol articles, which describe protocols hosted on protocols.io. Read more information on sharing protocols at https://plos.org/protocols?utm_medium=editorial-email&utm_source=authorletters&utm_campaign=protocols.

We look forward to receiving your revised manuscript.

Kind regards,

Pablo Martin Rodriguez

Academic Editor

PLOS ONE

“The author(s) received no specific funding for this work. Author RFL is employed and partially funded by Google. The funder provided support in the form of salary for RFL and the publication fee for this article, but did not have any additional role in the study design, data collection and analysis, decision to publish, or preparation of the manuscript.”

“Author RFL is employed by Google. This does not alter RFL’s adherence to PLOS ONE policies on sharing data and materials. Google has no restrictions on this work.”

Reviewers' comments:

Reviewer's Responses to Questions

**Comments to the Author**

1. Is the manuscript technically sound, and do the data support the conclusions?

Reviewer #1: Yes

Reviewer #2: Yes

2. Has the statistical analysis been performed appropriately and rigorously? 

Reviewer #1: N/A

Reviewer #2: N/A

3. Have the authors made all data underlying the findings in their manuscript fully available?

Reviewer #1: Yes

Reviewer #2: Yes

4. Is the manuscript presented in an intelligible fashion and written in standard English?

Reviewer #1: Yes

Reviewer #2: Yes

5. Review Comments to the Author

Reviewer #1: In this paper, the author investigates various bounds (including tight bounds) for the median the scaled

Gamma distribution. The results and proofs are quite technical and the author is nonetheless precise in their

treatment. The weaknesses I see are that the review of the literature is quite lacking (there are many works

on very close topics such as asymptotic bounds for the gamma, Poisson and negative binomial distributions

that are not mentioned, even very recent ones. Including them could help motivate the results of the paper

a bit more and help the reader put the results in perspective. Also, the figures could benefit from a legend

or a clearer labelling of the different curves. Other than that, the conclusions and the proofs look accurate.

I found the paper to be well-written, easy to follow, and of interest to researchers working on asymptotic

bounds for the gamma, Poisson and negative binomial distributions.

My recommendation is to accept the paper if the author can address the weaknesses mentioned above.

Reviewer #2: I appreciated reading this paper which presents clearly new bounds for the median of a standard gamma distribution. The paper is well written and clear. The problem addressed by the article is interesting and the results seem to be relevant to probability theory and could be potentially useful. However, the paper needs minor revisions before it can be accepted for publication. Thus, should the author answer adequately to my comments and suggestions below, then I would recommend the paper for publication in Plos One.

6. PLOS authors have the option to publish the peer review history of their article (what does this mean?). If published, this will include your full peer review and any attached files.

Reviewer #1: No

Reviewer #2: No

---

## [Author Response · Author response to Decision Letter 0]

25 Jun 2023

Rebuttal Letter / Response to Reviewers

by Richard F. Lyon

PONE-D-22-34782

24 June 2023

I appreciate the positive reviews from both reviewers.  

I mostly made changes in line with their suggestions, and added a few more bits along the way to help clarify (such as the new section "Toward a proof" that might help others complete the work, and an example of a third-order rational function interpolation with low relative error).

Reviewer #1:

…The weaknesses I see are that the review of the literature is quite lacking (there are many works

on very close topics such as asymptotic bounds for the gamma, Poisson and negative binomial distributions

that are not mentioned, even very recent ones. Including them could help motivate the results of the paper

a bit more and help the reader put the results in perspective. 

I have added three references to very recent related works. But I don’t understand them well enough to say much about how related they are, so I just said, "In addition to the works mentioned above, there have been several more recent works on asymptotic properties and bounds for medians and other quantiles of gamma distributions and of the closely related Poisson and negative binomial (or Polya or Pascal) distributions, with a variety of interesting approaches \\cite{priore2022approximate, ouimet2023refined, pinelis2021monotonicity}.

… Also, the figures could benefit from a legend or a clearer labelling of the different curves. 

I found that Figure 2 had an extra curve (from a corollary) that should not have been there, and that confused the interpretation of some of the labels on the curves, so I simply removed that (it was a thin solid red curve in the original figure). Now Figure 2 seems more clear.

For Figures 1 and 3 the labels on the curves seem clear and unambiguous. 

That leaves Figure 4, which I agree was a bit confusing and crowded, with the (conjectured) interpolated bounds being described in the caption but not in the figure itself. So I managed to squeeze in a legend, and removed the 53% and 54% quantile curves that were in that area, leaving the presentation more symmetric between 48% and 52% and leaving more room for the legend.

Reviewer #2

…the paper needs minor revisions … my comments and suggestions below …

Most of the suggestions were of the form “might be better to add a full stop after the equation” or “might be better to add a comma after the equation”. I followed all of those, but I think I also added another comma or two. In addition, many were “might be better to add the symbol ⇒ before the equation and a full stop after the equation”, which I also agreed with and did. In each case I followed the typical LaTeX styling advice of separating the end punctuation from the math by a thin-space. Where I already had full stops after a few equations, I moved them away by a thin-space. Most of these changes are hard to spot in the diff, as the full stop or comma are too short for underlining to show up, and the blue color is easy to miss. But they’re there.

I enumerate responses to the numbered suggestions that were more than these.

These 3 I group and treat out of order:

1. p.1-9: γ should be defined.

5. p.2-10: Again, γ should be defined.

12. p.4-13: Again, γ should be defined.

In the abstract I added “(with $\\gamma$ being the Euler--Mascheroni constant)”. In the text, on p.2, I added “(with $\\gamma \\approx 0.5772157$ being the Euler--Mascheroni constant)” which includes its approximate value. Thank you for noticing this was needed. I think it does not need to be said again on p.4.

2. p.1-17: might be “...probability density function (PDF)...” instead of “...PDF...”.

I took the liberty to add a possessive, this way: “The gamma distribution's probability density function (PDF) is …”

3. p.1-19: might be x>0 instead of x≥0. Note that if k=1 and x=0, we have p1(0) = 0/0, which is an indeterminate form.

Yes, x>0. Done.

4. p.1-23: might be “... cumulative distribution function (CDF)...” instead of “...CDF... ”.

Done

9. p.4-2: the author uses p(k,x) to denote the PDF, but in p.1 the notation used is pk(x).

I switched to using the subscript k consistently for both the PDF p_k(x) and the CDF P_k(x).

17. p.5-18: might be helpful to clarify that ν(k) > L4(k) for all k > 0.

I added “for all k > 0” and also added some clarifying words to the start of the paragraph, “Since the denominator and the exponent are positive…”. And I removed the redundant comment that followed: “The derivation holds for all k > 0.”

31. p.7-27: might be “when k < 1” (as in the theorem statement) instead of for all positive k.

I fixed it the other way. The theorem statement now says for all positive k, as in the table of theorems to prove. I re-checked that the proof is valid over that full domain.

50. p.11-22: avoid acronym like RHS and LHS without say its respective meaning.

Reworded as “Dividing by the positive right-hand side gives the equivalent condition…”.

Other changes I made, which you may notice in the diffs:

I changed “The positivity constraint is met” to “The constraint is met” in one place where the constraint inequality is not a comparison to 0.

Where I had “rather a” I changed to “rather than a”.

I changed “which is completes the proof” to “this completes the proof” in the proof of Lemma 4.

In the Fig. 4 caption I used a code font (\\texttt) for the Matlab function name gammaincinv.

I think that’s all. I tried to restrain my tendency to keep tweaking.

---

## [Decision Letter · Decision Letter 1]

2 Jul 2023

Tight bounds for the median of a gamma distribution

PONE-D-22-34782R1

Dear Dr. Lyon,

We’re pleased to inform you that your manuscript has been judged scientifically suitable for publication and will be formally accepted for publication once it meets all outstanding technical requirements.

Kind regards,

Pablo Martin Rodriguez

Academic Editor

PLOS ONE

Additional Editor Comments (optional):

Reviewers' comments:

Reviewer's Responses to Questions

**Comments to the Author**

1. If the authors have adequately addressed your comments raised in a previous round of review and you feel that this manuscript is now acceptable for publication, you may indicate that here to bypass the “Comments to the Author” section, enter your conflict of interest statement in the “Confidential to Editor” section, and submit your "Accept" recommendation.

Reviewer #1: All comments have been addressed

Reviewer #2: All comments have been addressed

2. Is the manuscript technically sound, and do the data support the conclusions?

Reviewer #1: Yes

Reviewer #2: Yes

3. Has the statistical analysis been performed appropriately and rigorously? 

Reviewer #1: Yes

Reviewer #2: N/A

4. Have the authors made all data underlying the findings in their manuscript fully available?

Reviewer #1: Yes

Reviewer #2: Yes

5. Is the manuscript presented in an intelligible fashion and written in standard English?

Reviewer #1: Yes

Reviewer #2: Yes

6. Review Comments to the Author

Reviewer #1: After thorough consideration and addressing all mentioned points, I find this version acceptable. The paper can be accepted.

Reviewer #2: The former report pointed out some minor revisions. These has been corrected in the revised version of the authors. More generally, all of the modifications made by the authors (included those that were not requested) are satisfactory. Thus, I recommend the paper for publication in Plos One.

7. PLOS authors have the option to publish the peer review history of their article (what does this mean?). If published, this will include your full peer review and any attached files.

Reviewer #1: No

Reviewer #2: No

---

## [Editor Report · Acceptance letter]

29 Aug 2023

PONE-D-22-34782R1 

Tight bounds for the median of a gamma distribution 

Dear Dr. Lyon:

I'm pleased to inform you that your manuscript has been deemed suitable for publication in PLOS ONE. Congratulations! Your manuscript is now with our production department. 

Kind regards, 

on behalf of

Professor Pablo Martin Rodriguez 

Academic Editor

PLOS ONE